# 9-N-n-alkyl Berberine Derivatives: Hypoglycemic Activity Evaluation

**DOI:** 10.3390/pharmaceutics15010044

**Published:** 2022-12-22

**Authors:** Mikhail V. Khvostov, Elizaveta D. Gladkova, Sergey A. Borisov, Marina S. Fedotova, Nataliya A. Zhukova, Mariya K. Marenina, Yulia V. Meshkova, Nicolae Valutsa, Olga A. Luzina, Tatiana G. Tolstikova, Nariman F. Salakhutdinov

**Affiliations:** 1N. N. Vorozhtsov Novosibirsk Institute of Organic Chemistry, Siberian Branch of the Russian Academy of Sciences, 9, Akademika Lavrentieva Ave., 630090 Novosibirsk, Russia; 2V. Zelman Institute for Medicine and Psychology, Novosibirsk State University, Pirogova Str. 1, 630090 Novosibirsk, Russia; 3Department of Natural Sciences, Novosibirsk State University, Pirogova Str. 1, 630090 Novosibirsk, Russia

**Keywords:** berberine derivatives, metabolic syndrome, hypoglycemic activity, OGTT, agouti yellow mice, hepatic steatosis

## Abstract

Several novel 9-N-n-alkyl derivatives of berberine (C5, C7, C10, C12) were synthesized. They were analyzed in vitro and in vivo for their hypoglycemic activity. In vitro studies showed that the derivatives with shorter alkyl substitutes at concentrations ranging from 2.5 to 10 μM were able to stimulate glucose consumption by HepG2 cells more prominently than the derivatives with longer substitutes (C10 and C12). All compounds demonstrated a better effect compared to berberine. Their impact on cells’ viability also depended on the alkyl substitutes length, but in this case, C10 and C12 derivatives demonstrated the best results. A similar correlation was also found in the OGTT, where the C5 derivative demonstrated a pronounced hypoglycemic effect at a dose of 15 mg/kg and C12 was less effective. This compound was further investigated in C57BL/6^Ay^ mice for four weeks and was administered at a dose of 15 mg/kg. Pronounced effect of C12 on carbohydrate metabolism in mice was discovered: there was a decrease in fasting glucose levels and an increase in glucose tolerance in OGTT on the 14th and 28th days of the experiment. However, at the end of the experiment, signs of hepatosis exacerbation and an increase in the content of hepatic aminotransferases in blood were found.

## 1. Introduction

Hyperglycemia is a major symptom of a number of metabolic disorders, including recently epidemic type 2 diabetes [1]. Hyperglycemia in type 2 diabetes occurs when insulin-dependent tissues’ (skeletal muscle, adipose tissue and liver) insulin sensitivity is disrupted, which leads to a significant decrease in their glucose uptake and hyperinsulinemia. Based on the feedback principle, the pancreas produces more insulin to lower the blood glucose concentration, which in the later stages of the disease leads to depletion of the islets of Langerhans beta cells and insulin deficiency. Often, carbohydrate metabolism disorders occur simultaneously with disorders of lipid metabolism, characterized by increased low-density lipoproteins and triglycerides blood concentrations. The described metabolic changes result in the progression of atherosclerosis and the nervous system damage [2]. Complications are a major cause of death and high disability in patients with diabetes mellitus. In order to control glycemia and blood lipid levels, hypoglycemic drugs are necessary [3,4].

Currently, there is not a great variety of therapeutic agents that increase tissue sensitivity to insulin. Only metformin and thiazolidinediones have such an effect, while the main mechanism of action of other medications used for T2DM treatment is the increase of insulin secretion (Glucagon-like peptide-1 receptor agonists (GLP-1), Dipeptidyl peptidase-4 inhibitors (DPP-4), sulfonylurea derivatives), a decrease in glucose absorption in the gastrointestinal tract (acarbose) or increased glucose kidney excretion (sodium glucose co-transporter-2 (SGLT2) inhibitors) [5]. Previously, we reported the discovery of a compound with a pronounced effect of increasing tissue sensitivity to insulin in a series of isoquinoline alkaloid berberine derivatives. Compound SHE-196 (9-(hexylamino)-2,3-methylenedioxy-10-methoxyprotoberine chloride) demonstrated a significant reduction in the mice’s blood glucose concentration after single administration in the dose range from 5 to 15 mg/kg, which is much lower than the clinical dosage of the parent compound berberine (380 mg/kg). In obese T2DM mice (C57BL/6^Ay^ mice, AY mice), its administration for three weeks revealed the following changes: increased glucose tolerance in OGTT, decreased fasting insulin levels and increased sensitivity to insulin, and decreased body weight and interscapular fat including brown, which indicates its activity increase [6].

In this work, we synthesized a series of SHE-196 analogues with a variation in the length of the aliphatic substituent to study their hypoglycemic activity and to reveal the structure–activity relationship.

## 2. Materials and Methods

### 2.1. Chemistry

The structure of the product was determined using ^1^H and ^13^C NMR spectra. ^1^H and ^13^C NMR spectra were recorded on Bruker AV-400 spectrometers (Bruker, Billerica, MA, USA) at 400.13 MHz (^1^H) and 100.61 MHz (^13^C) in DMSO-d6. Mass spectra were recorded on a Bruker micrOTOFQ using electrospray ionization (ESI). Silica gel (60–200 mesh, Macherey–Nagel, Düren, Germany) was used for column chromatography; chloroform with gradient of ethanol was used as eluent. Berberine chloride hydrate (TCI Co. (Tokyo, Japan)) was used after oven drying at 95 °C for 5 h. Pentylamine, hexylamine, heptylamine, decylamine, and dodecylamine were purchased from Acros Organics (Geel, Belgium).

### 2.2. General Procedure for the Synthesis of Compounds **2a-e**

The previously described procedure [6] was used to obtain the target compounds. Berberine chloride **1** (1 mmol) and 5 mmol of alkylamine (pentylamine, hexylamine, heptylamine, decylamine and dodecylamine) were heated to 120–130 °C for 4–5 h without solvent. Then, the reaction mixture was cooled to temperature 20–23 °C and diluted with acetone. The precipitate obtained was filtered, washed with acetone and purified by column chromatography (silica gel, eluent–chloroform and ethanol gradient) to obtain the compound **2a-e**. For spectral data, see Appendix A.

### 2.3. Biological Experiments

#### 2.3.1. In Vitro Experiments

##### Cell Culture

HepG2 cells were obtained from the shared research facility “Vertebrate cell culture collection”, Institute of cytology RAS. Cells were routinely cultured in DMEM high-glucose (Servicebio, Wuhan, China) containing 10% FBS (Sigma-Aldrich, São Paulo, Brazil), 100 μg/mL streptomycin, 100 U/mL penicillin and 0.25 μg/mL amphotericin B (Sigma-Aldrich, Saint Louis, MO, USA) at 37 °C in a 5% CO_2_ incubator (NuAire, Inc., Plymouth, MN, USA). Cells were passaged when cell fusion was over 80%, about twice a week.

##### The Design of the Experiment on HepG2 Cells

Cells were seeded into 96-well plates (TPP, Trasadingen, Switzerland) at a density of 3 × 10^5^ cells/mL in DMEM high-glucose containing 10% FBS. After culturing for 24 h, the cells were washed with HBSS (Gibco, Paisley, UK) twice and the medium was replaced with serum-free low glucose (5.5 mM glucose) DMEM (Servicebio, Wuhan, China) with berberine, C-5, C-6, C-7, C-10, C-12 at various concentrations (2.5, 5, 10 μM) and metformin (1, 2.5, 5 mM) (CAS 1115-70-4 Acros Organics, Geel, Belgium). There were three-to-four replicate wells for each treatment. After 24 h, the supernatant part was collected for glucose consumption and lactate release assay, while cell viability was measured using MTT assay.

##### Glucose Consumption and Lactate Release Assay

Glucose levels were assayed using a kit (Vector-Best, Novosibirsk, Russia) which was based on the glucose oxidase method, using photometer Multiscan Ascent (Thermo Labsystems, Helsinki, Finland). Glucose consumption was calculated by subtracting the glucose concentration in the supernatant of the blank well (DMEM without cells) from the glucose concentration in the supernatant of the well with cells. Data are presented as percentage of maximum glucose consumption (5.5 mM) and normalized to the control (treated with vehicle) for each plate. Meanwhile, the concentration of lactate in the supernatant was also determined using a commercial kit (Vector-Best, Novosibirsk, Russia). Lactate release are presented as percentage of maximum value (11 mM) and normalized to the control. 

##### MTT Assay for Cell Viability

Solution of MTT reagent (Thiazolyl Blue Tetrazolium Bromide (Panreac AppliChem, Darmstadt, Germany)) (5 mg/mL) was added to medium in a 1:10 volume ratio, and incubated for 4 h at 37 °C in the CO_2_ incubator. Then, the supernatant was removed and precipitate was dissolved with DMSO (Reagent Component, Moscow, Russia) for determination of the optical density. Cell viability was calculated as percentage of the control group. 

#### 2.3.2. In Vivo Experiments

##### Animals

Male C57BL/6 and C57BL/6^Ay^ mice weighing 23–25 g and 28–32 g, respectively, were used. Animals source: SPF vivarium of the Institute of Cytology and Genetics SB RAS. The animals were housed with ad libitum access to water and feed. Humidity, temperature and 12/12 h light-and-dark cycle in vivarium were controlled. All experiments with animals were conducted in accordance with the Russian Federation’s laws, the Ministry of Health of the Russian Federation decree no. 199n of 4/01/2016, the European Parliament and European Union Council Directive 2010/63/EU of 22/09/2010 on the protection of animals used for scientific purposes. The experiment protocol was approved by the Ethics Committee of N.N. Vorozhtsov Institute of Organic Chemistry SB RAS (protocol no. P-01-04.2022-14).

##### The OGTT

For the screening purposes, C57BL/6 mice (*n* = 6) were used. AY mice (*n* = 6) were subjected to the test during the long-term experiment. Animals fasted for 12 h before the test. Mice in all groups were given oral glucose at a dose of 2.5 g/kg 30 min prior to blood glucose concentration measurement. Compounds **2a-e** and berberine were administered orally at a dose of 15 or 30 mg/kg in tween-water suspension 30 min prior to glucose administration. During AY mice experiment, metformin (MF, CAS 1115-70-4 Acros Organics, Geel, Belgium) at a dose of 250 mg/kg was used as the positive control. First OGTT in AY mice was conducted after 14 days of administration and all compounds were introduced 30 min prior to the glucose load through oral gavage. Second OGTT was performed on 28th day of the experiment and the last compounds’ introduction was a day prior to test. For all mice, blood samples were obtained from a tail incision. The following time points were used: 0 (before dosing), 30, 60, 90, and 120 min after the glucose load. The ONE TOUCH Select blood glucose meter (LIFESCAN Inc., Milpitas, CA, USA) was used for blood glucose concentration measurement. Tai’s model was used to calculate the area under the glycemic curve (AUC) [7].

##### The AY Mice Experiment Design

Body weight gain was facilitated by adding lard and cookies to standard chow ad libitum for 30 days. Body weight of 35 g was a minimum requirement for the mouse to be chosen for the further experiment. Selected animals were divided in the following groups (*n* = 6 in each): (1) vehicle (water + 2 drops of Tween 80), (2) **2e** 15 mg/kg, (3) MF 250 mg/kg, and 4) C57BL/6 mice (*n* = 6) + vehicle. The diet was the same during the whole experiment. Tested compounds were given orally (through oral gavage) once a day. OGTT was conducted on the 14th and 28th day of the experiment. Animals were decapitated and blood was drawn for the biochemical assay after 31 days of the experiment. For the histology following tissues were taken: liver, interscapular white and brown fat, pancreas. In C57Bl/6 mice, for histological analysis, gonadal white fat was taken instead of interscapular. Body weight and food consumption were evaluated once a week.

##### Biochemical Assays

Serum was separated using centrifugation at 1640× *g* for 15 min. Standard diagnostic kits (Vector-Best, Novosibirsk, Russia) and a photometer Multiscan Ascent (Thermo Labsystems, Helsinki, Finland) were used to analyze serum total cholesterol, triglycerides, alkaline phosphatase, alanine aminotransferase and lactate levels.

##### Histological Examination

Excised tissues (liver, interscapular white and brown fat, pancreas) were fixed in 10% neutral buffered formalin for 7 days; then, they were subjected to the standard dehydration in ascending ethanol concentrations and xylene. Tissue samples were embedded in paraffin on an AP 280 workstation using Histoplast (Thermo Fisher Scientific, Waltham, MA, USA, melting point of 58 °C). Slices with a thickness of 4.5 μm were obtained on a rotational microtome NM 335E with disposable interchangeable blades. Periodic acid–Schiff, hematoxylin and eosin, and orange G staining were used. Sample’s examination was done under a light microscope at a magnification of ×100–400.

##### Statistical Analysis

Statistical analysis was performed using the Mann–Whitney U test and one-way ANOVA followed by the Fisher LSD test for multiple comparisons. Data are shown as mean ± SEM. Data with *p* < 0.05 were considered statistically significant.

## 3. Results

### 3.1. Synthesis

First, 9-N-n-alkylberberine derivatives **2a-e** with a variation in the length of the aliphatic substituent from C5 to C12 (Figure 1) were obtained in one step by heating berberine chloride and 1-alkylamine without solvent using an adapted [8]. Compounds **2a-e** were obtained with yields 27–52% after purification by column chromatography.

### 3.2. In Vitro

In the in vitro experiments, it was found out that all new derivatives at concentrations of 2.5–10 μM increased glucose consumption and lactate release in HepG2 cells (Figure 1A). The dependence of the effect on the structure of the compounds is clearly seen: derivatives with a shorter aliphatic fragment length have a more pronounced activity. Compounds **2a** (C-5), **2b** (SHE-196; C-6) and **2c** (C-7) at a concentration of 2.5 μM already increased glucose consumption to the level of 87–97%, whereas compounds **2d** (C-10) and **2e** (C-12) increased it to 54–65%. The initial berberine 1 was inferior to all derivatives in the potency of its action. Metformin was used as a positive control in higher concentrations (1–5 mM), which were chosen bases on the literature data [9]. Metformin increased glucose consumption comparably with berberine. The effect of the studied compounds on lactate release was similar (Figure 1B).

The most pronounced increase in glucose consumption and lactate release led to a decrease in cell viability: at concentrations of 2.5 μM for compound **2a** (C-5) and **2b** (C-6), 5 μM for **2c** (C-7) and 10 μM for **2d** (C-10) (Table 1). At the same time, compound **2e** (C-12) had almost no effect on cell viability over the entire concentration range.

### 3.3. OGTT Screening

OGTT results showed a similar result to the in vitro data on structure–activity relationship in berberine derivatives. It was found that the increase in the length of the aliphatic fragment resulted in a decrease in the hypoglycemic action and lethality of mice from severe hypoglycemia. Moreover, this trend was observed at both used doses (15 and 30 mg/kg) (Table 2). The initial berberine 1 at a dose of 30 mg/kg showed no effect. According to the combined in vitro and OGTT data, of all the studied substances for further long-term administration to AY mice at a dose of 15 mg/kg, compound **2e** (C12) was chosen, the pharmacological effect of which was accompanied by the highest safety profile.

### 3.4. Body Weight and Food Consumption of AY Mice

Throughout the experiment on AY mice, the dynamics of changes in their body weight and the amount of feed consumed were being noted. A decrease in body weight in compound **2e** mice was observed right after the first week of administration and it further intensified by the end of the experiment. After 4 weeks of administration, mice in this group did not differ much from the C57Bl/6 mice in body weight. Weight loss was also observed in metformin-treated mice, but its severity was significantly lower (Figure 2). The dynamics of changes in mice body mass correlated with the level of feed intake in all groups except the AY control mice, which had a decrease in feed intake while still gaining weight in the fourth week of the experiment (Table 3).

### 3.5. OGTT after 14 Days of Experiment

A significant hypoglycemic effect of compound **2e**, greater than that after a single injection, was detected in the OGTT after two weeks of the experiment. At the same time, the fasting glucose level in mice of this group was lower than in mice receiving metformin (Figure 3).

### 3.6. OGTT after 28 Days of Experiment

After 4 weeks of the experiment, the fasting glucose level was lower in mice receiving compound **2e** and MF. The OGTT was performed without the substances’ administration 30 min before the glucose load (the last administration of substances was 24 days before that), so the data obtained reflect the cumulative effect of the agents’ administration. As can be seen from Figure 4, the hypoglycemic effect of compound **2e** in this test was significantly higher than that of metformin.

### 3.7. Biochemical Blood Test

The biochemical parameters’ study of mice blood revealed a significant increase in liver transaminases and lactate levels, while a decrease in triglyceride levels was observed (Table 4).

### 3.8. Weight of Organs and Tissues of AY Mice

At the end of the experiment, the liver, gonadal and interlobular fat were taken from all animals and their weight was determined. The mass of adipose tissue and liver of mice receiving compound **2e** differed significantly from the animals of the negative control. The liver weight of mice in group **2e** had a large variation in values, which was reflected in the average value, which was almost one and a half times higher than in AY mice, but had no statistical validity. At the same time, the body weight of these mice was significantly lower than that of the AY group. The weight of adipose tissue, on the contrary, was lower (Table 5).

### 3.9. Histology

Histological examination demonstrated degenerative and necrotic changes in the AY mice’s liver (Figure 5). All these findings indicate the development of fatty hepatosis in AY mice. In the exocrine part of the pancreas, no dystrophy or necrosis were detected. In the endocrine part, pronounced hyperplasia of islet apparatus was detected (Figure 6). Brown adipose tissue analysis revealed a marked fat content increase in adipocytes. Large fat droplets merged with each other to form fat cysts (Figure 7). White adipose tissue examination discovered a prominent increase in the adipocytes’ size and their fusion into fat cysts (Figure 8). Metformin-treated AY mice showed improvement in the described metabolic abnormalities (Figure 5). In the exocrine part of the pancreas, focal fatty dystrophy of acinocytes was detected. In the endocrine part, the diameter of the islet apparatus was decreased (Figure 6). Metformin administration resulted in fat content decrease in brown and white adipose tissue: the reduction of fat’s droplet size was observed (Figure 7 and Figure 8). In AY mice treated with compound **2e**, there was an increase in liver metabolic disorders. In these animals, the development of total fatty hepatosis was observed (Figure 5 and Figure 9). No alterations (dystrophies, necrosis) were revealed in the exocrine part of the pancreas. In the endocrine part, pronounced hyperplasia of the islet apparatus was observed similar to mice from the negative control group (Figure 6). Reduced fat content was observed in brown adipose tissue. Adipocytes were dominated by mainly small fat droplets (Figure 7). The size of adipocytes in the white adipose tissue also slightly decreased, indicating the activation of lipolysis processes (Figure 8). For more detailed histological description, see Appendix A.

## 4. Discussion

Several 9-N-n-alkylberberine derivatives **2a-e** with a variation in the length of the aliphatic substituent from C5 to C12 were obtained in one step from berberine chloride **1** and 1-alkylamine using an adapted procedure [8] for the synthesis of N-substituted berberine derivatives. N-n-pentyl-substituted derivative **2a** is described in the literature as an antibacterial agent with properties against *Staphylococcus aureus* [10]. All other synthesized compounds are new and were described by NMR, IR and MS spectra (Appendix A).

Firstly, the synthesized berberine derivatives were studied in vitro, where their ability to influence the glucose consumption and lactate release of HepG2 cells was evaluated. This cell line was used due to being identical to normal hepatic cells physiological principles of glucose metabolism [11]. This experiment revealed the dependence of glucose consumption and lactate release expression on the length of the alkyl fragment in 9-N-n-alkylberberine derivatives. These processes diminished with its increase, i.e., the strongest glucose consumption and lactate release were observed in the group of compound **2a** and the weakest effect in the group of compound **2e**. At the same time, cell survival had another dependence, i.e., compound **2e** was the safest for the cells. The initial berberine 1 in all concentrations studied was inferior to all derivatives in terms of the effect’s severity. The most obvious reason for the increased glucose consumption by HepG2 cells under the action of the studied substances can be considered the increase of AMP-activated protein kinase a1 (AMPKa1) activity, previously shown for berberine in similar experiments [12].

Subsequent studies of these substances in OGTT at doses of 15 and 30 mg/kg have clearly demonstrated that the effect, which was observed on cells, is also fully manifested in animals. An increase in the methylene groups number in the alkyl fragment of the berberine’s 9-N-derivative led to a loss of hypoglycemic activity, which subsequently required a dose increase. Berberine itself in similar doses showed no hypoglycemic effect (Table 2), which is a consequence of its low [13]. The derivative with the shortest alkyl fragment (compound **2a**) had the most pronounced hypoglycemic effect on animals among all the studied substances, which caused the death of several mice in the corresponding group. The analysis of the data obtained in vitro and in the OGTT, as well as the results of earlier experiments on AY mice with the compound **2b** (SHE-196) [6], suggested that it would be interesting to study the pharmacological action of the compound, which was the least active in the OGTT, but sufficiently effective in vitro, in the hope of a cumulative effect from its long-term administration while maintaining a low risk of fatal hypoglycemia. Therefore, compound **2e,** in this case, was the lead compound for administration to AY mice at a dose of 15 mg/kg for four weeks. These mice demonstrate antagonism toward melanocortin receptors by the agouti protein, which is the result of the agouti gene (Ay/a) mutation. In mice, it causes yellow pigmentation, late-onset obesity, and hyperinsulinemia [14]. These metabolic changes make AY mice a convenient animal model for studying hypoglycemic effects and related impact on lipid metabolism.

As early as 1 week after the beginning of the experiment, mice receiving compound **2e** developed a trend of body weight loss. This trend persisted until the end of the experiment, with the mass of the mentioned mice closely approaching that of C57Bl/6 mice (Figure 2). A similar influence on body weight dynamics was previously observed in the case of compound **2b** (SHE-196) when administered to AY mice, but the decrease was less significant [6]. The reduction in body weight corresponded to a decrease in the feed intake of these animals. Body weight also had negative dynamics in the metformin group, which was expected, since it is known for being able to reduce body weight and lipid synthesis [15] and has been previously noted in our work [6].

We began to evaluate the hypoglycemic effect of tested agents after 14 days of their administration in the OGTT. Judging by the initial fasting blood glucose measures it was already safe to say that the hypoglycemic action of compound **2e** was more pronounced than that of MF. At the same time, the response of mice to the glucose load after the administration of both compound **2e** and MF was the same, indicating the accumulation of compound **2e**’s effect, as we did not observe such an effect after a single gavage of compound **2e** (Figure 3). After four weeks of compound **2e** administration, the fasting glucose levels of these mice were similar to those of the mice receiving MF. Significant differences were found in the OGTT; in this case, substances were not administered 30 min before glucose (the last injection was the day before the test), and their cumulative effect was evaluated. Compound **2e** showed a much stronger effect, which is clearly visible in the correspondent glycemic curve and AUC figures (Figure 4). In addition, the administration of **2e** led to a decrease in the mass of interlobular adipose tissue, both white and brown, which usually indicates the activation of lipolysis [16]. However, a concerning factor was a significant increase in liver weight, which, along with its creamy color at autopsy, indicated an aggravation of its damage, which could be due to fatty hepatosis, ever-present in AY mice, [17] or some other lesion. This alteration of the liver was also reflected in the biochemical analysis of the blood. Significant increases in transaminases and lactate were found, while TG levels were lower than in AY control mice (Table 4). These changes indicate the presence of hepatocyte cytolysis and the inability of the liver to carry out gluconeogenesis [18]. This was confirmed by the histomorphological analysis of liver sections from mice treated with compound **2e**. Total fatty hepatosis and massive cytolysis were detected, which can be considered an aggravation of the metabolic disorders occurring in AY control mice, since no signs of toxic liver damage, such as necrosis, were observed. On the contrary, the derivative with a shorter alkyl fragment **2b** (SHE-196) studied in our previous work, also on AY mice, led to the resolution of fatty liver dystrophy [6]. Here it is worth noting that berberine itself in doses of 50 to 150 mg/kg is capable of causing liver damage, and specifically in animals with diabetes [19].

The structure of the fatty interlobular tissue reflected the reduction in its mass—there was a decrease in lipid droplets, which can be regarded as an intensification of lipolysis and activation of the brown fat [20]. The administration of compound **2e** had no effect on the pancreatic islet apparatus, unlike metformin, which reduced its hypertrophy and the studied earlier compound **2b** [6]. We can conclude that the cumulative effect hypothesis of the least active derivative in OGTT screening was confirmed in the experiment performed on AY mice. Compound **2e** was found to have prominent hypoglycemic effect. However, the detected increase in metabolic damage to the liver suggests that there is probably no point in further lengthening the aliphatic fragment in 9-N-n-berberberine derivatives synthesized as hypoglycemic agents.

## 5. Conclusions

Four 9-N-n-alkyl derivatives of berberine (**2a-e**) with different lengths of the aliphatic substituent (from C-5 to C-12) were synthesized in the present work. The previously synthesized and studied 9-N-n-hexyl derivative of berberine [6] (formerly no in vitro data were obtained for it) was additionally studied for its structure–activity in vitro. As a result of the experiments on glucose consumption with HepG2 cells and OGTT data, it was found that increasing the length of the alkyl fragment in these berberine derivatives leads to a decrease in the severity of hypoglycemic action with an increase in their safety. The derivative with the weakest effect in OGTT but a pronounced effect in vitro and the best indexes of cell and mouse survival from hypoglycemia, the 9-N-dodecyl derivative of berberine **2e**, was studied in more detail when administered to AY mice (obese and DM2 mice) for four weeks. It was found that the administration of compound **2e** to these animals resulted in a significant hypoglycemic effect and a marked improvement in glucose tolerance. However, in addition to this, it was observed that this substance can aggravate the fatty hepatosis in AY mice. Thus, we can conclude that further lengthening of the alkyl fragment of such derivatives is not promising in regard to the studied biological activity.

## Data Availability

Not applicable.

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
