# Peer review of "9-N-n-alkyl Berberine Derivatives: Hypoglycemic Activity Evaluation"

_pharmaceutics, 2022, doi:10.3390/pharmaceutics15010044_

Round 1

Reviewer 1 Report

The reviewer did appreciate the quality of the manuscript and the interesting research performed.

To further improve the report, the reviewer suggest the authors to integrate the Metformin values in the figures 1 A & B beside the control bar for the in vitro graph.

Author Response

Thank you very much for your comments, which helped us improve the quality of our manuscript. Our detailed responses are provided below.

The reviewer did appreciate the quality of the manuscript and the interesting research performed.

To further improve the report, the reviewer suggest the authors to integrate the Metformin values in the figures 1 A & B beside the control bar for the in vitro graph.

Response: Metformin values added into figures 1A and 1B.

Reviewer 2 Report

This is an interesting manuscript regarding potential new antidiabetic agents. I have some concerns and suggestions for its improvement.

1. The manuscript is too long.

2. I suggest shortening the abstract, reducing information to only relevant details for this study. Also, GLP-1 receptor agonists should be mentioned, as weight reduction is important in reducing insulin resistance as well as berberine derivatives (compound 2E) particularly, as described.  

3. It is necessary for Materials and Methods section to be shortened. I strongly suggest the compound description to be moved to the Supplementary material. 

4. The Results section is so overwhelmed with information, that it is confusing. I also strongly suggest presenting only the most relevant results, which are mentioned in Conclusion, while putting other results into the Supplementary material. 

5. Discussion is too explanatory, but necessary in this way. Nevertheless, I am not sure if the conclusion is sound. If it turned out that the compound 2e is toxic, it is necessary to explain, where, anyway, its utility might be of value.

Author Response

Thank you very much for your comments, which helped us improve the quality of our manuscript. Our detailed responses are provided below.

This is an interesting manuscript regarding potential new antidiabetic agents. I have some concerns and suggestions for its improvement.

  1. The manuscript is too long.

Response: We have moved chemistry description and full histological results (text) to the Supplementary materials, so the article’s length was reduced.  

  1. I suggest shortening the abstract, reducing information to only relevant details for this study. Also, GLP-1 receptor agonists should be mentioned, as weight reduction is important in reducing insulin resistance as well as berberine derivatives (compound 2E) particularly, as described.  

Response: The abstract was shortened. Glucagon-like peptide-1 receptor agonists are added into the introduction.

  1. It is necessary for Materials and Methods section to be shortened. I strongly suggest the compound description to be moved to the Supplementary material. 

Response: Compounds description was moved to Supplementary material.

  1. The Results section is so overwhelmed with information, that it is confusing. I also strongly suggest presenting only the most relevant results, which are mentioned in Conclusion, while putting other results into the Supplementary material. 

Response: We believe that all presented results are important to better understand the pharmacological properties of tested compounds. We agree that histological results part may be too complex for the readers, so we have shortened it and its original version was moved to the Supplementary material

  1. Discussion is too explanatory, but necessary in this way. Nevertheless, I am not sure if the conclusion is sound. If it turned out that the compound 2e is toxic, it is necessary to explain, where, anyway, its utility might be of value.

Response: the present work is a continuation of our study of 9-N-substituted berberine derivatives and is intended to show how changing the length of the alkyl substituent affects the hypoglycemic properties of the molecule. The results show that the substance 2e shows additional effects leading to increased fatty hepatosis in animals, which may be a consequence of its effect on targets that are not affected by shorter derivatives or the targets are similar, but the affinity of 2e is more pronounced. So far, the most logical explanation for the results obtained seems to be that there is no need to study derivatives as hypoglycemic agents with a substituent larger than C12 and to focus on the modification of derivatives with shorter substituents.

Round 2

Reviewer 2 Report

i have no further remarks.